# Meta-Transfer-Learning-Based Multimodal Human Pose Estimation for Lower Limbs

**DOI:** 10.3390/s25051613

**Published:** 2025-03-06

**Authors:** Guoming Du, Haiqi Zhu, Zhen Ding, Hong Huang, Xiaofeng Bie, Feng Jiang

**Affiliations:** 1School of Computer Science and Technology, Harbin Institute of Technology, Harbin 150001, China; 19b903025@stu.hit.edu.cn; 2School of Medicine and Health, Harbin Institute of Technology, Harbin 150001, China; haiqizhu@hit.edu.cn; 3College of Computer and Control Engineering, Northeast Forestry University, Harbin 150040, China; dingzhen@nefu.edu.cn; 4School of Computer Science and Engineering, Sichuan University of Science & Engineering, Zigong 643002, China; huanghong@suse.edu.cn; 5Unmanned Systems Technology Research Institute, Northwestern Polytechnical University, Xi’an 710072, China; biexiaofeng@nwpu.edu.cn

**Keywords:** human pose estimation, sEMG, multimodal, transfer learning, meta learning, knowledge fusion

## Abstract

Accurate and reliable human pose estimation (HPE) is essential in interactive systems, particularly for applications requiring personalized adaptation, such as controlling cooperative robots and wearable exoskeletons, especially for healthcare monitoring equipment. However, continuously maintaining diverse datasets and frequently updating models for individual adaptation are both resource intensive and time-consuming. To address these challenges, we propose a meta-transfer learning framework that integrates multimodal inputs, including high-frequency surface electromyography (sEMG), visual-inertial odometry (VIO), and high-precision image data. This framework improves both accuracy and stability through a knowledge fusion strategy, resolving the data alignment issue, ensuring seamless integration of different modalities. To further enhance adaptability, we introduce a training and adaptation framework with few-shot learning, facilitating efficient updating of encoders and decoders for dynamic feature adjustment in real-time applications. Experimental results demonstrate that our framework provides accurate, high-frequency pose estimations, particularly for intra-subject adaptation. Our approach enables efficient adaptation to new individuals with only a few new samples, providing an effective solution for personalized motion analysis with minimal data.

## 1. Introduction

Accurate HPE remains a crucial task in the field of human–machine interaction (HMI), with accuracy and stability being pivotal for user-centric applications. Sensing technologies such as imaging, sEMG, and inertial measurement unit (IMU) have facilitated advancements in human motion representation, and when integrated with estimation methods and neural networks, have driven continuous advancements in human motion analysis and estimation methodologies [1,2,3,4,5,6,7,8,9,10,11,12,13,14,15,16,17,18,19]. As HMI applications have become increasingly prevalent in daily life, the diversity in motion patterns introduces significant complexity and uncertainty to processing models. While intra-subject adaptation remains essential for personalized HMI, its implementation introduces compounded challenges: designing context-aware architectures, curating specialized motion datasets, and preserving estimation consistency. These interdependent barriers fundamentally undermine pose estimation reliability. Knowledge-based approaches, such as transfer learning (TL) and meta-learning (ML), have proven effective in enhancing performance, particularly for intra-subject scenarios [20,21,22,23,24,25]. Additionally, leveraging multimodal information integration and effective knowledge fusion can further enhance the performance of HPE tasks [26,27,28,29,30,31,32,33,34].

Traditional HPE approaches leverage high-frequency and sensitive sEMG signals for motion state estimation and partial body posture analysis, such as hand gesture recognition [6], classification [7,8], and upper and lower body movement estimation [9,10]. Furthermore, sEMG-based methods demonstrate significant potential in motion analysis by utilizing encoding networks such as BP [11], DBN [12], CNN [13], LSTM [8], and hybrid architectures [14]. Given the complexity of motion states in irregular daily movements, vision-based methods exhibit higher reliability and accuracy due to the lack of initial value corrections and instability in slow or stationary states. Previous studies employing traditional machine learning and deep learning techniques [15,16] have offered vision-based solutions for HPE tasks. Vision-based solutions like AplhaPose [17], Openpose [18] and MotionBert [19] demonstrate strong potential in practical applications. However, they often lack the robustness and reliability required for demanding tasks, such as precise operational interactions. The incorporation of multi-view capturing [35] and multi-level feature extraction [36] further enhances estimation accuracy. However, vision-based methods typically demand extensive network parameters and substantial computational resources, potentially causing latency in real-time interactive applications. Consequently, human pose prediction and estimation have shifted toward multimodal processing approaches. Significant latency reduction has been achieved by integrating IMU and video streams [37,38,39], while the fusion of RGB and depth information [40,41] yields more reliable spatial results. To achieve both accuracy and efficiency in HPE tasks, RGB and sEMG data are utilized as our baseline modalities. However, in the context of body movement analysis, the translation and reshaping of the human body occur simultaneously. Inspired by previous work of rigid body pose estimation using VIO information [42,43], we use the pelvis as the central reference point for estimating the relative pose during movement. By incorporating VIO information as a pose movement descriptor, the representation of human motion is further enriched.

Despite significant progress in HPE, many previous studies fail to ensure robustness and reliability when pre-trained models are applied to subjects with varying motion patterns, habits, or physical characteristics. To address this challenge, online optimization [22] and adaption [23,24] techniques have shown great promise in real-time motion processing. By incorporating neural networks into the adaptation process [25] these techniques further enhance the accuracy of the results. However, sensor noise and shifts in data distribution in the input stream can compromise the stability and robustness of online computations. To improve generalization and resistance to interference, larger and more diverse training datasets, along with deeper networks, have been employed in pose estimation [44,45,46] and activity recognition [47,48] tasks leveraging TL techniques based on IMUs and video streams. Few-shot learning technologies such as ML have been widely applied in computer vision [49], natural language processing [50] and recommender systems [51], providing an effective solution for limited training data and uncertain scenarios. Meta-transfer learning techniques [52,53] offer advantages in adaptability across various tasks and scenarios, leveraging a knowledge-based migration framework. Knowledge-based learning algorithms have demonstrated strong potential in human motion prediction [49,54] and classification [55], particularly in handling uncertainty in domains and tasks with limited reference movements. Given the diversity of motion types and patterns, knowledge-based mechanisms provide higher-dimensional descriptive parameters in prediction and estimation networks, ensuring the reliability and robustness of human motion modeling.

Contemporary human pose estimation (HPE) systems demonstrate marked performance advantages through multimodal fusion architectures [26,27], contrasting with the inherent limitations of unimodal approaches. This paradigm enhances system robustness via three synergistic mechanisms: (1) cross-modal feature complementarity that mitigates sensory noise through differential error profiles, (2) hierarchical representation learning enabling discriminative kinematic pattern recognition, and (3) dynamic attention weighting optimized for intra-subject adaptation. Particularly in personalized motion analysis contexts, the fused feature manifold provides physiologically consistent representations that substantially improve movement decoding fidelity under real-world variability conditions. To further enhance stability and efficiency in knowledge-based meta-learning for human motion, multimodal information fusion combined with knowledge-level techniques has shown promise in training and inference. Early-stage fusion methods like bagging [28] and embedding [29] offer a solid foundation for improving model performance, while intermediate fusion strategies, such as feature-level fusion, enable synergy between complementary modalities. Vision-based multimodal deep learning networks [30,31,32] and attention-based networks [33,34] are effective at leveraging feature extraction to enhance results, particularly in intra-subject scenarios. However, relying solely on data-level or feature-level fusion does not ensure effective knowledge transfer between tasks. Overcoming this limitation requires deeper exploration of advanced knowledge representation techniques. Techniques like shared feature spaces [56], shared representations [57], and shared model parameters [58] facilitate knowledge transfer across domains, ensuring relevant information is preserved and applied. These methods facilitate knowledge transfer, improving the generalization capabilities of models trained on multimodal data. Despite these advancements, misalignment between information and knowledge across domains remains a key challenge. Aligning data representations, particularly when source and target domains differ significantly, remains a major challenge in intra-subject adaptation. Addressing this misalignment demands sophisticated fusion techniques and refined adaptation strategies to reconcile discrepancies in knowledge transfer across new contexts.

Based on extensive advancements in HPE using multimodal sensing technologies, the remaining task of dealing intra-subject adaptation still being challenging. In this paper, we propose a novel framework for multimodal information with knowledge fusion aiming to improve both accuracy and stability in HPE tasks. We summarized our main contributions as follows: (1) a framework for multimodal input knowledge fusion in meta-transfer learning, enhancing accuracy and stability; (2) a multi-channel feature extraction and fusion network designed to improve knowledge representation capabilities, and also resolve the knowledge alignment problem; and (3) a training and adaptation framework incorporating few-shot learning for the efficient updating of encoders and decoders, enabling dynamic feature updating in real-time applications.

## 2. Materials and Methods

### 2.1. Equipment and Participants

As no public dataset meets our requirements (shown in Table 1) for multimodal information, we recruited volunteers to perform ground walking movements while being equipped with four types of sensors: (1) an RGB camera (OmniVision, Shanghai, China) was used to capture visual motion data for the modulation of bone movements; (2) six sEMG sensors were placed on the rectus femoris (RF), biceps femoris (BF), and gastrocnemius (GA) of both legs to capture muscle activity; (3) an Intel RealSense T265 (Intel, Santa Clara, CA, USA) sensor, mounted on the waist, was used to capture body motion; and (4) a Vicon motion capture system (Vicon, Denver, CO, USA) was employed to obtain ground-truth movement data for the lower body. The placement of all sensors and capture environment are illustrated in Figure 1, and the specifications of all sensors are provided in Table 2. The selected RGB module offers robust dynamic motion capture stability, making it well-suited for motion analysis scenarios. The Vicon capture system, equipped with T40 cameras, provides high-resolution skeleton spatial references, while Delsys sEMG signal capture sensors (Delsys, Natick, MA, USA) offer high capture frequency with minimal errors. Both systems are widely utilized in human motion analysis research. The Intel RealSense T265 camera directly provides integrated VIO information, which helps conserve computational resources.

Ten healthy volunteers (ten males; age: 23–26 years, average: 24.2 ± 1.8 years, height: 160–185 cm) were recruited in the experiment. All participants had no history of joint or neurological disorders or injuries that could limit their exercise ability. This study was conducted in accordance with the Declaration of Helsinki. Written informed consent was obtained from all participants.

### 2.2. Data Acquisition

Each participant was required to perform 12 walking trials on flat ground, consisting of 4 trials at slow speed, 4 trials at moderate speed, and 4 trials at fast speed. Each trial lasted 1 min, followed by a 1 min pause to prevent muscle fatigue. Prior to the experiment, each participant was instructed to walk on a treadmill with a zero incline at speeds of 3.5 km/h, 4.5 km/h, and 6 km/h to familiarize themselves with slow, moderate, and fast walking speeds. The walking trials were conducted in a circular path with two possible directions: clockwise and counterclockwise, with each trial following only one direction.

The Vicon motion capture system employs reflective markers to obtain the 3D spatial coordinates of the markers, with joint locations calculated based on methods outlined in prior work [66,67]. The system, which consists of 10 infrared cameras, is calibrated using a “T”-shaped tool with infrared LEDs on its surface. The calibration process is rigorously executed using built-in software (Nexus v2.16), within a defined action zone measuring 3 m by 4 m. To address capture frequency discrepancies across multimodal equipment, we use hard-wired synchronization between the sEMG signals and the Vicon capture system. The VIO and RGB streams were timestamped and stored using the same clock as the sEMG capture system.

### 2.3. Method

In addressing the optimization challenges of pose estimation adaptation for intra-subject scenarios, we implemented a multi-stage meta-transfer learning strategy. This approach effectively reduces reliance on large-scale pre-training datasets while significantly conserving computational resources required for model training and validation. To enhance the reliability and stability of estimation outputs, we developed customized knowledge fusion mechanisms that strategically integrate multimodal input streams through modality-specific feature integration protocols. As shown in Figure 2, the proposed meta-transfer learning framework consists of three stages, taking known and new data as inputs to the learning process. The learning process includes a pre-training stage, which generates an initial pose estimation model with pre-trained weights and biases. The meta-transfer learning stage fine-tunes the pre-trained model on multiple tasks by modulating its parameters. Finally, in the meta-adaptation stage, the meta-learner adapts to new tasks using a few examples beyond the known sample set and is evaluated on test data.

#### 2.3.1. Pre-Training

To effectively address the knowledge alignment challenge, the pre-training stage encodes multimodal inputs separately, enabling a reliable and robust fusion of features through knowledge sharing. Figure 3 illustrates the complete pre-training process, with the final output comprising the six joint angles of lower limbs (hip, knee, ankle of both legs), describing the movements.

Given input stream as $I^{m}\in R^{T_{m}\times D_{m}}$, where m∈i,e,v that represents the input of image stream, sEMG stream and VIO stream, respectively. $T_{m}$ and $D_{m}$ denote the input length and channel size. As shown in Figure 4, the convolutional block attention module (CBAM) [68] leverages excellent representation of channel attention and spatial attention or effective feature representation. An attention block is applied to the image stream input, while a transformer captures temporal dependencies in the sEMG and VIO streams. A CBAM-Resnet12 block consists of four residual blocks, each connected to three convolutional layers, a CBAM layer, and a max-pooling layer. The feature extraction process is defined as $r^{m}=X^{*}\left(I^{m}\right)$, where $X^{*}$ represents the CBAM-Resnet12 or transformer, depending on the input stream.

The inherent differences in representational capacity among multimodal information streams for human pose estimation often result in input misalignment. The primary objective of knowledge sharing is to mitigate these discrepancies by ensuring feature alignment across individual clips. To address this challenge, we perform the following transformation on the encoded features:(1)$${\tilde{r}}_{m}=f^{m}\left(r_{m}\right),$$
where ${\tilde{r}}_{m}$ denotes the transformed feature representation. The training process of the transform encoder involves evaluating the relationship between the target feature dimension and the encoded feature dimension. To measure the discrepancy between these feature spaces, we utilize the Euclidean distance as the loss function, which ensures the alignment of features during the optimization process.(2)$$L_{KS}=\sum_{\forall (k,t)\in M,k\neq t}^{M}m^{m}EU\;\left(k,\;t\right).$$
where $m^{m}$ represents for trainable normalization function for each modality. Therefore, the remaining low-level features enhance the learning performance during the following process.

To ensure feature coherence across varying spatial and temporal scales, the fusion network is designed with three branches. Each branch consists of a different number of blocks, where each block comprises a convolutional layer, batch normalization, and a leaky ReLU activation function. The outputs from these branches are concatenated through global average pooling, producing the fused feature k. The decoder processes the encoded representations of joint angles through a series of operations, including convolutional layers, batch normalization, ReLU activations, and dropout. Finally, a fully connected regression layer generates the final predictions. The loss function employed for training is the mean squared error (MSE), defined as:(3)$$L_{PE}=\frac{1}{N}\sum_{i=1}^{N}{\left(y_{i}-{\hat{y}}_{i}\right)}^{2},$$
where $y_{i}$ represents the ground-truth values, ${\hat{y}}_{i}$ denotes the predictions, and N denotes the number of joint angles.

#### 2.3.2. Meta-Transfer Learning

With a pre-trained pose estimation model initialized with parameters (W,b), the meta-learning process utilizes a parameter modulation strategy to fine-tune the model for task-specific adaptation. This strategy ensures that the model can effectively generalize to new tasks by optimizing the parameter space based on task-specific objectives. The process can be formally expressed as:(4)$$ML\left(W,b\right)=W\gamma^{w}+\left(b+\gamma^{b}\right),$$
where $\gamma^{w}$ and $\gamma^{b}$ denote modulation parameter to pre-trained model.

In the meta-transfer stage, the pre-trained model is fine-tuned using new tasks and corresponding samples to enhance its task-specific performance. To achieve this, gradient-based update techniques are employed for parameter modulation, enabling the model to adapt efficiently to the new task distribution. For an N-way K-shot task T, the training loss function is defined as:(5)$$L_{T}=\frac{1}{N\times K}\sum_{y_{i}\in T}^{T}L_{MSE}(y_{i}).$$
where $L_{MSE}$ represents the MSE loss calculation and $y_{i}$ is the given sample. The optimized pose estimation model is evaluated using query sets of size N×Q, where N represents the number of classes and Q denotes the number of queries per class, sampled from the same task distribution. By utilizing the full support set and query set, the meta-learner is iteratively trained and fine-tuned to achieve an adaptation-ready state. This approach ensures robust generalization across task variations, enabling the model to effectively handle new, unseen scenarios with minimal additional training.

#### 2.3.3. Meta Adaptation

In this stage, the meta-learner is tasked with processing samples that differ from those used during training, allowing it to demonstrate its ability to quickly adapt to novel tasks. Randomly selected samples from the support set are employed to fine-tune the task-specific model, where the meta-learner iteratively adjusts its parameters to align with the requirements of the new task. The adaptation performance is then quantitatively evaluated by calculating the average accuracy on the query set. This evaluation provides a comprehensive assessment of the meta-learner’s capacity to generalize effectively across tasks, ensuring robust performance in diverse application scenarios. The meta-learning framework utilizes task-specific fine-tuning with minimal data from the target individual, allowing the model to capture individual-specific features while preserving its generalized knowledge. Theoretically, this framework is designed to handle multiple new tasks during the adaptation stage. However, since the primary objective of the proposed method is to optimize personalized adaptation, the experiments in this study specifically focus on single-person tasks.

### 2.4. Evaluation

As the intra-subject adaptation is the main target of proposed method, we define the estimation of each subject as a specific task in meta-learning process and a ten-second clip of input is defined as a one-shot. Additionally, walking phase (stance phase and swing phase) and walking speed are also defined as a task for some experiments to further evaluating online interactive capabilities of proposed method in diverse applicational scenarios.

To define the accuracy of human poses, we introduce root mean square error (RMSE) to evaluate the testing results. RMSE calculated the distance between estimation values and ground-truth values, and the ground-truth values are calculated by labeled data captured by the Vicon motion capture system.

## 3. Results

The primary objective of the proposed method is to achieve effective pose estimation adaptation for intra-subject scenarios. To evaluate its performance, we tested the method on different subjects using pre-trained models of varying scales. Pre-trained models were generated using 50%, 75%, and 100% of the remaining data, representing small-scale, middle-scale, and large-scale models, respectively. The performance of the pre-trained models was compared to that of the proposed method across different subjects. Visual results in Figure 5 demonstrate significant improvement achieved through the 1-way 5-shot meta-learning process. The reduction in estimation error achieved through this process is 37.4%, 24.7%, and 23.8% for the small-scale, middle-scale, and large-scale pre-trained models, respectively. Moreover, the improvement observed in meta-learning with the small-scale pre-trained model surpasses that of the large-scale pre-trained model, which presents a trade-off strategy between training resources and meta-learning samples. The estimation errors for each subject are not identical; however, by leveraging knowledge adaptation through the meta-transfer learning process, the differences among subjects in few-shot scenarios are reduced compared to the pre-trained model.

To evaluate the contributions of multimodal inputs to the proposed method, we conducted an ablation study focusing on input modalities. Since VIO information lacks spatial context for pose estimation, it was evaluated in combination with other modalities. Results presented in Table 3 demonstrate that incorporating additional modalities enhances performance in both pre-trained models and meta-learning scenarios. The estimation errors of RGB information are more reliable in terms of spatial accuracy compared to sEMG data; however, when sEMG signals encode continuous motion information, performance improves. Notably, in walking scenarios, when VIO information is combined, the estimation errors decrease significantly by leveraging the richer descriptive information of the human body.

The goal of this research is to optimize real-time applications by implementing intra-subject adaptation using multimodal information, which may introduce computational complexities. We conducted an ablation study on inference efficiency across different modalities, with the results presented in Table 4. RGB information incurs a higher cost in both inference time and memory due to the complexity of the CBAM-ResNet12 encoding network. However, as shown in Table 3, spatial information significantly impacts accuracy. While sEMG and VIO information are higher frequency compared to RGB data, the computational resource requirements are not significantly large relative to their contribution, benefiting from knowledge sharing and the meta-transfer learning strategy. Overall, the proposed framework achieves real-time performance for less intense movements, making it suitable for target scenarios such as healthcare monitoring and cooperative robotics. From the perspective of memory cost, the proposed method also demonstrates strong potential for end-device implementation in low-cost applications. For further optimization of inference efficiency, the encoding network for RGB information can be replaced with a less complex sequence-based network, without significant loss in accuracy, due to the knowledge transfer capabilities of the proposed method.

We conducted an ablation study to thoroughly investigate the pose estimation network utilizing multimodal knowledge fusion. In this study, the data fusion module in the proposed method was replaced with a concatenation operation to integrate features from different modalities. Table 5 presents the differences between knowledge-sharing (KS) and non-knowledge-sharing (nKS) processes. The results indicate that the KS module significantly enhances performance in both pre-trained models and meta-learning scenarios. Specifically, in nKS cases, even 1-shot learning demonstrates the ability to extract sufficient knowledge from the provided sample queries. This finding highlights that without the KS module, simple feature concatenation does not efficiently contribute to achieving high-accuracy pose estimation.

Variations in walking speeds result in distinct movement patterns, while different walking phases produce unique joint spatial distributions. We evaluated the knowledge transfer capabilities for tasks involving different walking speeds and walking phases. As shown in Table 6 and Table 7, the number of learning shots significantly impacts performance, exhibiting varying effects depending on the task type.

For different walking speed tasks, we observe that the estimation error decreases as the number of learning shots increases. Notably, the rate of error reduction is higher for fast walking compared to moderate and slow speeds, indicating that the proposed method excels at extracting knowledge from the distinct movement patterns associated with intense activities. For different walking phases, a similar trend is observed. In the stance phase, performance improves as the number of learning shots increases. However, in the swing phase, the error decreases more rapidly than in the stance phase, attributed to the smoother joint movements during this phase. In these two cases, we obtained convincing evidence that the proposed method enhances pose estimation results in both relatively slow and rapid motion scenarios. Furthermore, in slow-motion cases, estimation errors are smaller due to the spatial information encoded in the RGB data.

In HMI systems, specific tasks are often closely associated with the movements of particular joint. In our study, the seven analyzed joints exhibit distinct movement patterns, offering valuable insights for understanding other types of motion. As illustrated in Figure 6, the estimation errors for both legs are nearly symmetrical, with only slight variations. The hip estimation error is the lowest, attributed to its relatively smaller range of motion and the effective representation provided by VIO features. The knee and ankle joints exhibit higher estimation errors, although they undergo significant optimization through meta-transfer learning. The ankle joint, which is challenging to estimate due to its agile movement patterns, achieves comparable estimation error levels to other joints through multimodal pose estimation and meta-transfer learning. Notably, our quantitative analysis revealed distinct optimization patterns following meta-transfer learning: knee and ankle joint estimation errors decreased significantly compared to hip joint errors. This phenomenon stems from biomechanical characteristics where knee and ankle joints exhibit greater intra-subject variability due to their higher degrees of freedom, which existing pre-trained models inadequately capture given limited training data diversity. The meta-learning framework effectively addressed this limitation through subject-specific motion pattern adaptation, particularly benefiting from the sequential encoding of continuous periodic movements. Crucially, the differential optimization efficacy across joints correlates with their kinematic profiles—while knee and ankle primarily demonstrate sagittal plane motion during gait cycles, the hip’s multiplanar movement complexity presents greater adaptation challenges.

Due to the lack of a public dataset that satisfies our data requirements, baseline methods were adapted and evaluated using our custom dataset. As shown in Table 8, the proposed method demonstrates significant advantages over baseline approaches on our dataset, which include sEMG-based transfer methods [69,70,71] and an RGB information-based pre-trained model [19]. MyoNet, EMGNet and other EMG-based transfer learning use sEMG signals for estimating joint angles through transfer learning methods, employing convolutional networks and deep learning techniques. In contrast, the proposed method integrates RGB and VIO modalities effectively within the meta-transfer learning process. Large-scale pre-trained models, such as MotinBert, have achieved excellent results in estimating human pose from RGB data. However, inference on 2D images lacks the dimensional constraints required for 3D pose estimation, leading to relatively higher estimation errors.

## 4. Discussion

In this study, the primary objective is to optimize pose estimation adaptation for intra-subject cases. By incorporating few-shot learning in the meta-adaptation stage, general knowledge is effectively transferred to the meta-learner, enabling it to develop a comprehensive global understanding of new tasks, represented in this study by the new target. In practical applications, interactive systems and equipment are required to respond to instructions with precise adjustments. The meta-adaptation mechanism facilitates this requirement by enabling accurate and rapid adaptation to dynamic inputs. Comparisons with baseline methods, including pre-trained and transfer-learning-based models, demonstrate that the proposed method successfully fulfills the task and outperforms alternative approaches.

Since the performance of an inference model is closely tied to the scale of the training dataset, we conducted a deeper investigation into subject-level cases. We observed that the number of learning shots has a similar impact on reducing prediction error as the data scale used during pre-training. From a statistical perspective (Figure 5), few-shot learning demonstrates an “amplifier” effect on the pre-trained estimation model. This indicates that for any subject, the estimation performance of a smaller pre-trained model can match that of a larger pre-trained model with the assistance of the meta-learner. Considering general knowledge transfer, the samples provided to the meta-learner serve as personalized contextual information, enabling the pre-trained model to adapt. This process logically aligns with the transition from a known knowledge domain to a filtered new knowledge domain. The adaptation process for each subject exhibits a consistent trend, further validating the stability and robustness of the proposed method.

Multimodal input plays a crucial role in enhancing the accuracy and robustness of HPE. In the proposed method, RGB data are utilized as a relatively precise spatial representation of the skeleton model, while sEMG signals capture continuity and muscle activity. Although VIO information is not commonly employed in HPE tasks, it offers inherent advantages in reducing noise and representing holistic movement patterns. Integrating VIO information into feature encoding significantly reduces estimation error, particularly in intense movement patterns. The proposed method seamlessly integrates these input sources without compromising the flexibility to decouple modalities when necessary. Furthermore, the method offers adaptability to various application requirements, allowing trade-offs between accuracy, stability, and frequency by selecting appropriate input modalities.

Different input modalities are often configured in distinct ways to capture diverse information distributions. During continuous motion, each modality responds uniquely to the same movement process or patterns. Direct feature fusion and knowledge extraction can introduce collateral noise, negatively affecting the performance of the pre-trained model. To ensure high-quality knowledge for training the meta-learner, a separately trained knowledge transfer module is employed, significantly enhancing the effectiveness of adaptation.

Understanding precise motion in the context of common task requirements is essential for achieving accurate HPE. In this study, movement patterns are categorized by walking speed and walking phases, with the efficiency of knowledge transfer closely linked to the intensity of these patterns. This indicates that the quantity and quality of learning samples significantly influence movements, particularly in cases where joints exhibit a wider range of motion. From the perspective of partial movement, each joint exhibits distinct movement patterns throughout the walking procedure. In this context, whether the goal is to balance estimation errors or to improve the performance of specific joint estimations, augmenting the pre-trained dataset or enriching learning samples can be beneficial.

The proposed method has shown promising results in intra-subject adaptation for HPE tasks. However, for broader applications, more complex and non-repetitive scenarios have not been addressed, indicating that further challenges remain. In future work, we aim to explore several key areas to further enhance the proposed framework. One direction involves investigating the impact of random and unpredictable movements on the performance of the model, particularly in scenarios where consistent patterns are absent. Additionally, we will examine the intricate relationship between knowledge transfer and feature representation, especially in the context of refined and nuanced movement patterns. Another critical area of focus will be the study of partial body movement patterns, assessing how localized movements influence overall pose estimation accuracy and the potential for improving adaptability in tasks requiring incomplete or partial input data. These efforts aim to expand the framework’s applicability and robustness across diverse and dynamic real-world scenarios.

## 5. Conclusions

This study proposed a comprehensive framework for multimodal input knowledge fusion within meta-transfer learning, significantly enhancing the accuracy and stability of human pose estimation. By integrating diverse modalities such as sEMG, VIO, and image data, the framework addresses uncertainties in complex tasks and compensates for the limitations of single-modal information. A multi-channel feature extraction and fusion network was developed to improve knowledge representation capabilities and resolve knowledge alignment challenges, ensuring robust and generalized performance across various input modalities. Additionally, a training and adaptation framework incorporating few-shot learning enables efficient encoder and decoder updates, achieving real-time dynamic feature adaptation. These innovations demonstrate strong potential for real-world applications requiring rapid adaptation and personalization, such as healthcare monitoring and human–machine interaction, offering enhanced precision, improved robustness, and scalable adaptability across diverse scenarios.

## Figures and Tables

**Figure 1 sensors-25-01613-f001:**
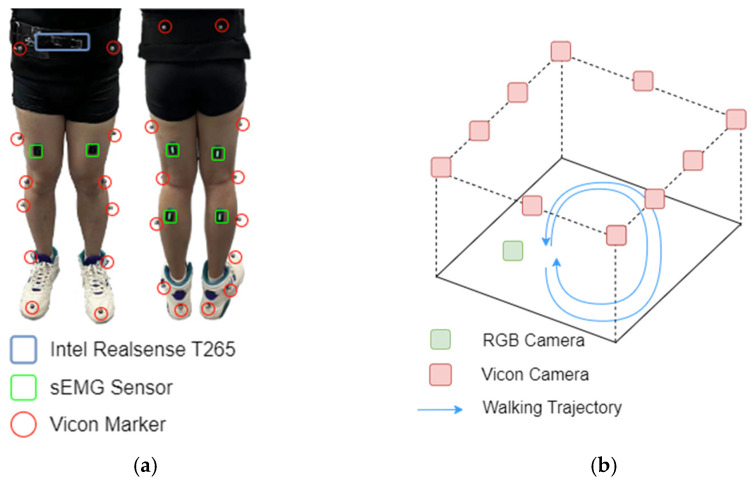
Sensor placement and data collection environment: (**a**) For the lower body, six sEMG sensors were placed on both sides of the legs, while 16 Vicon markers were used to collect ground-truth data. An Intel RealSense T265 sensor was mounted on the waist. (**b**) Ten Vicon cameras were positioned on the ceiling to capture reflective markers on the lower body, and an RGB camera was placed on the side wall. The subject performed walking trials on flat ground, both clockwise and counterclockwise.

**Figure 2 sensors-25-01613-f002:**
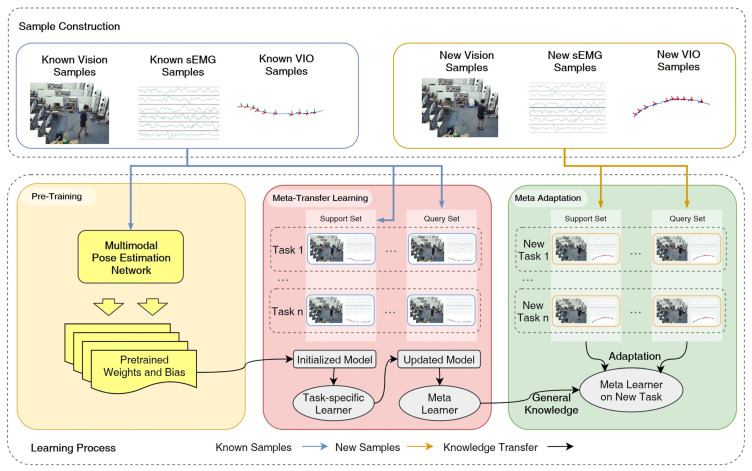
Overall schematic of proposed framework, totally including three phases.

**Figure 3 sensors-25-01613-f003:**
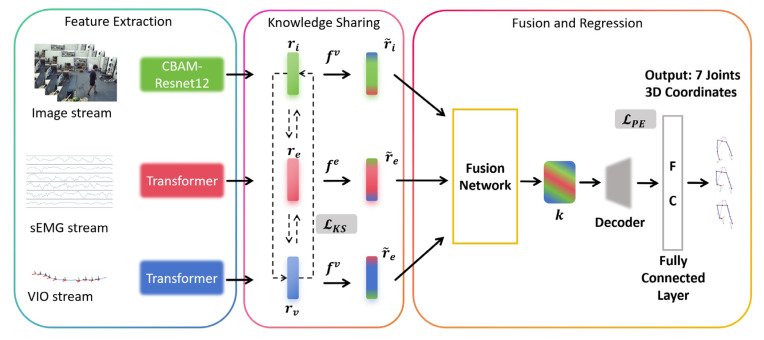
The pose estimation network is pipelined with feature extraction, knowledge sharing, fusion of knowledge and pose regression.

**Figure 4 sensors-25-01613-f004:**
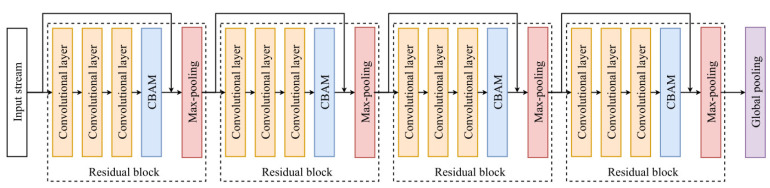
The structure of CBAM-Resnet12 is composed of a combination of CBAM module, residual block, convolution layer and max pooling layer.

**Figure 5 sensors-25-01613-f005:**
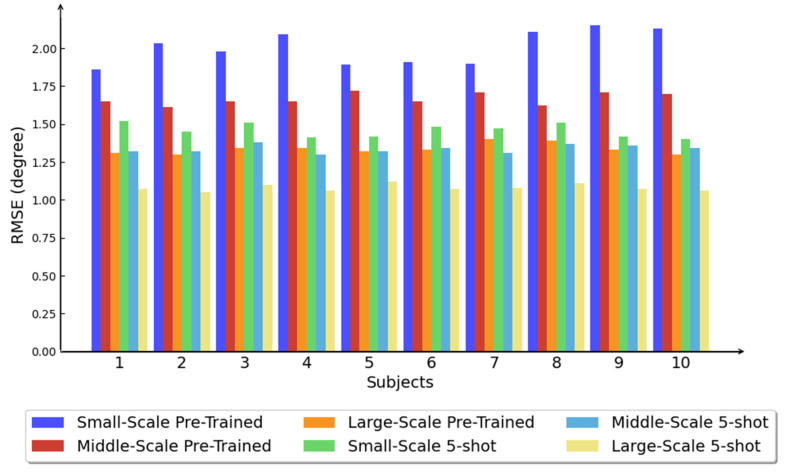
Results on different subjects with different scales of pre-training.

**Figure 6 sensors-25-01613-f006:**
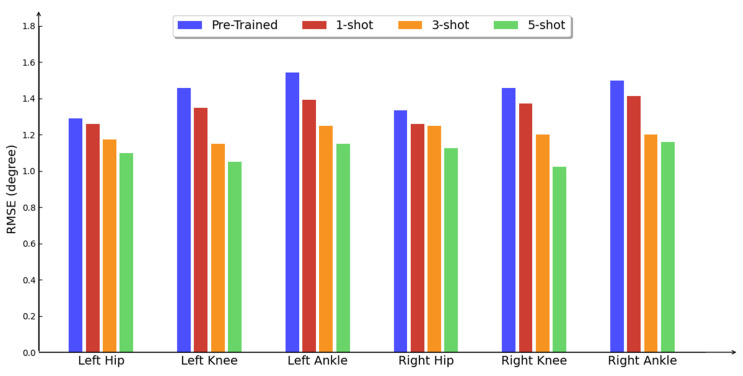
Evaluation of different joints from lower body, results are calculated with RMSE in degrees.

**Table 1 sensors-25-01613-t001:** Comparison between public dataset with ours.

Dataset	Modalities	Location
Requirements	RGB + VIO + sEMG	lower body
UCI dataset [59]	sEMG	upper arms; upper legs
COCO-WholeBody [60]	RGB	body
Human3.6M [61]	RGB	body
Action 3D [62]	Depth	body
ENABL3S [63]	sEMG + IMU	lower limbs
HAR-sEMG [64]	sEMG	lower limbs
Ninapro 7 [65]	sEMG + IMU	forearm
Ours	RGB + VIO + sEMG	lower body

**Table 2 sensors-25-01613-t002:** Specifications of sensors used in experiments.

Sensors	Modality	Resolution	Frame Rate (Hz)	Sensor Number	Sensitivity
RGB camera *	RGB	640 × 480	30	1	8-bit
Vicon T40	Skeleton	2336 × 1728	100	10	10-bit
Delsys Tringo	sEMG	-	1111.111	6	16-bit
Intel RealSense T265	VIO	-	200	1	16-bit

* The RGB camera is an industrial product with model HF877; in cases where purchase is not possible, we provide the sensing chip model OV 9750 for reproducibility of the research.

**Table 3 sensors-25-01613-t003:** Ablation study on input modalities, results are the estimation errors with RMSE in degrees in few-shot cases.

Vision	sEMG	VIO	Pre-Trained	1-Shot	5-Shot
√			1.67	1.59	1.43
	√		1.74	1.61	1.50
√	√		1.38	1.33	1.23
√		√	1.45	1.42	1.31
	√	√	1.54	1.43	1.38
√	√	√	1.37	1.30	1.07

The symbol √ represents the usage of input modalities.

**Table 4 sensors-25-01613-t004:** Ablation study on input modalities, the experiments are conducted on a single NVIDIA RTX 4080 SUPER GPU (Nvidia, Santa Clara, CA, USA).

Modalities	Inference Time (ms)	Inference Memory (MB)
RGB	13	847
sEMG	6	416
RGB + VIO	18	1278
sEMG + VIO	9	827
RGB + sEMG	21	1303
RGB + sEMG + VIO	24	1852

**Table 5 sensors-25-01613-t005:** Ablation study on KS and nKS cases, results the estimation errors with RMSE in degrees in few-shot cases.

Cases	Pre-Trained	1-Shot	5-Shot
KS	1.72	1.34	1.03
nKS	2.41	1.55	1.20

**Table 6 sensors-25-01613-t006:** Evaluation on different walking speeds, estimation errors are with RMSE calculated in degrees.

Walking Speed	Pre-Trained	1-Shot	3-Shot	5-Shot
Slow (3.5 km/h)	1.30	1.26	1.17	1.04
Moderate (4.5 km/h)	1.39	1.33	1.22	1.12
Fast (6 km/h)	1.44	1.36	1.28	1.24

**Table 7 sensors-25-01613-t007:** Evaluation on different walking phases, estimation errors are calculated with RMSE in degrees.

Walking Phase	Pre-Trained	1-Shot	3-Shot	5-Shot
Stance Phase	1.30	1.25	1.15	1.03
Swing Phase	1.41	1.33	1.22	1.15

**Table 8 sensors-25-01613-t008:** Comparisons of RMSE in degrees between different methods over different tasks.

	MyoNet [69]	EMGNet [70]	TL-EMG [71]	AplhaPose [19]	Ours	Ours
RMSE	1.88 *	1.97 *	2.62 *	1.52 *	1.30 **	1.07 ***

* Model is pre-trained transfer model which is not few-shot testable. ** Results of 1-shot learning. *** Results of 5-shot learning.

## Data Availability

The data are available from the corresponding author on reasonable request.

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
