# Peer review of "Meta-Transfer-Learning-Based Multimodal Human Pose Estimation for Lower Limbs"

_sensors, 2025, doi:10.3390/s25051613_

Round 1
Reviewer 1 Report
Comments and Suggestions for Authors
The basis for the selection of experimental equipment may be added as appropriate;
Motivation for the research: solution and design of the existing problem.
Discussion of results: for the results of the experiments may be further discussed as to why the differences exist. Limitations of the methodology of this paper and future perspectives can be pointed out.
References can be added to include more recent studies related to multimodal fusion and meta-migration learning to better demonstrate the research background and related work of this paper.
There is room for optimizing the graph format.
Reviewer 2 Report
Comments and Suggestions for Authors
The paper proposes a meta-transfer learning framework for multimodal human pose estimation (HPE) of lower limbs, integrating sEMG, VIO, and RGB data for accurate and stable pose analysis. The framework leverages few-shot learning to enable rapid adaptation to new subjects with minimal data, addressing challenges in intra-subject variability and alignment of multimodal inputs. Experimental results demonstrate significant improvements in pose estimation accuracy and robustness, outperforming baseline models and reducing errors across diverse tasks such as walking speeds and phases. The approach is particularly effective in healthcare and human-machine interaction scenarios, offering scalable and efficient solutions for real-time applications.
Here are some of my questions or suggestions:
1 In the first introduction paragraph, please use a group of references to support some general statements. For example, in Line 47, 49 in page 2.
Some of suggested reference about lower-limb human pose estimation papers:
https://doi.org/10.1016/0020-7101(95)01161-7
https://doi.org/10.1016/B978-0-12-815659-9.00004-4
https://doi.org/10.1016/j.knosys.2024.111810
https://doi.org/10.1109/ROBIO.2012.6491251
https://doi.org/10.1109/EMBC46164.2021.9630765
Also please consider citing some popular studies in general human pose estimation area, such as openpose.
2 In the second section, I might not agree with this statement that 'As no public dataset meets our requirements for multimodal information,'. Please make a comprehensice review of currents studies with type of sensors. A table would be suitable.
3 In equation 7, the full stop should be replaced with comma.
4 No need to introduce the existing studies with figure, such as the structure of CBAM.
5 In the experimental results section, please add and discuss more about baseline methods.
Reviewer 3 Report
Comments and Suggestions for Authors
Main Comment:
This paper proposes a multimodal lower limb posture estimation framework based on meta-transfer learning, aiming to improve the accuracy and stability of human posture estimation in interactive systems. Existing methods have limitations when addressing individual differences and complex movement patterns. Therefore, this study combines multiple sensor data, including surface electromyography signals, visual-inertial odometry, and high-precision images, to enhance model accuracy and stability through knowledge fusion strategies. Additionally, a few-shot learning training framework is employed, enabling the encoder and decoder to efficiently adapt and update. The research team recruited 10 healthy volunteers, used various sensors for data collection, and trained the model in three stages: pre-training, meta-transfer learning, and meta-adaptation. Experimental results demonstrate that the framework reduces estimation errors on pretrained models of different scales; furthermore, it performs well across various gait speeds and stage tasks, outperforming traditional baseline methods.
Overall, the paper is clearly expressed, logically consistent, and the writing style and method design are credible. The models used by the authors' research team show good performance on the self-built dataset. However, I noticed that the experimental content is not comprehensive, and I have concerns about the adequacy of some analyses. I will detail my points of concern in the following sections.
Specific Comments:
1. Lack of Coherence in Related Work: When elaborating on the research background, the analysis of the development status and challenges of multimodal information fusion in the field of human pose estimation, and the use of VIO information as a pose motion descriptor, failed to emphasize the importance of the current research in addressing accuracy and stability. It should echo the analysis of the misalignment between cross-domain information and knowledge, similar to the knowledge alignment strategy proposed in this paper.
2. Incomplete Description of Experimental Details: In the data collection section, specific parameters and calibration methods of the Vicon motion capture system were not detailed, which may affect the reproducibility of the research. Some figures and data lack necessary explanations. For example, in the different joint error evaluation figure (Fig. 6), the reasons for the error distribution differences were not fully analyzed. The knee and ankle joints showed higher estimation errors, and the authors achieved estimation error levels comparable to other joints using multimodal pose estimation and meta-transfer learning. The reasons for this distribution difference could be further explained.
3. Content of Figures and Tables: The format of some key information in the figures and tables was not detailed. For example, in Figure 1, there are three types of arrow lines; what is the difference between them? In Figures 2/3, the description of CBAM-Resnet12 is not detailed enough to show where the CBAM module is inserted. Table 6 lacks unit annotations.
4. Limitations of the Dataset: Relying on a self-made dataset and lacking comparison with public datasets, it would be beneficial to include results on datasets such as the UCI dataset or COCO-WholeBody dataset.
5. Lack of Ablation Study on Model Complexity and Efficiency: Although the network structure with the added CBAM module improves performance, it may increase computational costs. The paper does not analyze the impact on real-time applications, such as inference time and memory usage.
Reviewer 4 Report
Comments and Suggestions for Authors
This paper explores the importance of accurate and reliable human pose estimation in interactive systems, especially for application scenarios that require personalized adaptation, such as controlling collaborative robots, wearable exoskeletons, and medical monitoring devices. An innovative meta transfer learning framework is proposed to address the resource consumption and time cost issues associated with continuously maintaining diverse datasets and frequently updating individual adaptation models. This framework integrates multimodal inputs such as high-frequency surface electromyography (sEMG), visual cortex odometry (VIO), and high-precision image data, significantly improving the accuracy and stability of pose estimation through knowledge fusion strategies. At the same time, the author also solved the problem of multimodal data alignment, ensuring seamless integration between different modes. Introduced a training and adaptive framework with minimal shot learning ability. This design enables the encoder and decoder to efficiently update in real-time applications, achieve dynamic feature adjustment, and effectively respond to individual differences and scene changes. This framework can provide accurate high-frequency pose estimation, especially in terms of internal adaptation of subjects. At the same time, only a small number of new samples are needed to effectively adapt to new individuals, providing a practical and feasible solution for personalized motion analysis with minimal data.
In terms of experimental design, due to the inability to find a public dataset that meets the requirements of multimodal information, volunteers were recruited to conduct ground walking motion experiments and equipped with four types of sensors to comprehensively capture motion data. The experimental design fully considers the complexity of practical applications, and ensures the accuracy and reliability of data through carefully designed sensor placement and capture environment.
This article proposes a feature transformation strategy to address the issue of input misalignment caused by inherent differences in representation capabilities between multimodal information flows. Through this strategy, the problem of feature alignment has been effectively alleviated, further improving the accuracy of pose estimation. The writing of this article is relatively standardized, the logic is clear, the experimental design is reasonable, and the results are credible. I agree to publish it.
Round 2
Reviewer 2 Report
Comments and Suggestions for Authors
I have no further concern.
Reviewer 3 Report
Comments and Suggestions for Authors
I have thoroughly evaluated the author's revisions in response to my comments and found them to be comprehensive and satisfactory. I am pleased to recommend the acceptance of this paper in its current form.